# Physical Attributes of an Irrigated Oxisol after Brassicas Crops under No-Tillage System

José Luiz Rodrigues Torres [1,*], José Carlos Mazetto Júnior [2], Bruna de Souza Silveira [2], Arcângelo Loss [3], Gilsonley Lopes dos Santos [4], Renato Lara de Assis [5], Ernane Miranda Lemes [2] and Dinamar Márcia da Silva Vieira [2]

[1] Federal Institute of Triângulo Mineiro, Campus Uberaba, St. João Batista Ribeiro, n° 4000, Uberaba 38064-790, MG, Brazil

[2] Agricultural Sciences Institute, Federal University of Uberlândia, Campus Glória, Road BR 050, km 78, Uberlândia 38410-337, MG, Brazil; jcmazettojr@hotmail.com (J.C.M.J.); brunasilveira@iftm.edu.br (B.d.S.S.); ernanefito@gmail.com (E.M.L.); marcinha_0202@hotmail.com (D.M.d.S.V.)

[3] Center of Agricultural Sciences, Federal University of Santa Catarina, Itacorubi, Florianópolis 88034-000, SC, Brazil; arcangelo.loss@ufsc.br

[4] Postgraduate Program in Environmental and Forest Sciences, Federal Rural University of Rio de Janeiro, Seropédica 23890-000, RJ, Brazil; leylopes85@hotmail.com

[5] Federal Institute Goiano, Campus Iporá, Iporá 76200-000, GO, Brazil; relassis@bol.com.br

\* Correspondence: jlrtorres@iftm.edu.br

**Abstract:** In no-tillage areas, cover crops are a continuous supply of organic matter and other positive improvements to the soil's structural quality. We evaluated soil physical attributes in area cultivated with brassica crops on residues of cover crops cultivated under no-tillage. Six cover crops coverages [1-Brachiaria (B), 2-Sunn hemp (S), 3-Pearl millet (M), 4-S + B; 5-B + M; 6-S + M] and a native area (Cerrado biome), were evaluated for soil resistance to penetration (RP), soil density (SD), soil macroporosity, microporosity, volumetric moisture (VM), weighted mean diameter, geometric mean diameter, and aggregate stability index. RP and VM differed among treatments; no compacted soil layer was observed at up to 0.4 m soil depth; Low RP and SD were observed for Brachiaria and Pearl millet (Poaceae) compared to Sunn hemp (Fabaceae) in deeper soil layers; The principal components and cluster analysis indicated B + M as the most promising coverage for deep soil structuring. The soil physical quality indicators showed that millet in isolated cultivation or intercropped with another cover was the culture that presented the best results for most of the evaluated characteristics. The best indices of soil aggregation were observed where the species of the Poaceae Family were being cultivated in isolation or intercropped with each other.

**Keywords:** soil porosity; soil resistance to penetration; soil aggregation; winter crops

## 1. Introduction

In the past 50 years, the no-tillage system (NTS) of soil management has been consolidated as one of the most conservative systems for sustainable agricultural production in the Brazilian Cerrado. The low soil revolving, the maintenance of crop residues on the soil surface, and an efficient crop rotation system cause positive changes to the soil's physical, chemical, and biological attributes [1]. The continuous supply of organic residues to the soil surface, associated with the positive action of the roots from the cultivated crops, alter the soil structural quality and benefits the following crops, favoring crop root development and improving crop yield [2–4].

The soil's physical attributes are indicators of changes to soil quality. The most used soil attributes are soil resistance to root penetration, soil bulk density, macroporosity, microporosity, total soil porosity, soil water retention capacity, and stability of aggregates [5–7]. These attributes reflect the quality and state of conservation of the soil, enabling the identification of areas with nutrient deficiencies and soil physical constraints caused by the

movement of agricultural machines in the area [8]. Refs. [9,10] highlighted that these soil attributes are a low-cost alternative to efficiently diagnose the soil's physical quality.

In areas cultivated with grains under NTS, problems with soil compaction have aggravated due to the use of heavy and more efficient agricultural machinery, which changes the dynamics of air and water flow in the soil, hindering gas exchanges between the soil and the atmosphere and the plant development [11]. In areas cultivated with vegetables, the traffic of heavy machinery is lower. However, it still occurs during crop management in NTS areas or during the soil incorporation of crop residues and buildup of the seedbeds in conventional tillage areas [12,13].

When there is a need to decompress soil layers, the operation is usually done with chisel plows, rippers, and plows, but the effects caused by this soil mobilization tends to disappear due to the natural soil reconsolidation, which occurs due to the cycles of wetting and drying, and the traffic of agricultural machinery during crop management [14].

The plant species of Fabaceae and Poaceae, among others, are widely cultivated prior to commercial crops for the production of straw for ground cover, which cycle considerable amounts of nutrients to protect the soil against erosive processes, maintain its moisture, which help in the structuring of the soil, increase the stability of aggregates and help in the management of soil compaction in cultivation areas. The root system of these plants can cycle nutrients, grow in compacted soil layers, form stable biopores, improve soil water infiltration, and aggregate the soil particles [3,15]. However, few studies have been conducted in areas cultivated with vegetable crops.

The evaluation of soil physical attributes in irrigated areas under no-tillage is necessary because there is a continuous addition of organic matter. In conditions of high temperatures and humidity, the soil's organic matter provides favorable conditions for increasing the biodiversity of microorganisms in the soil [16]. These microorganisms change all the processes that involve the soil organic matter dynamics [17], accelerating the changes to the physical attributes [2], which need to be better evaluated. In this context, this study's objective was to assess soil physical quality in cultivated areas with different cover crops preceding the cultivation of vegetables in the irrigated area.

## 2. Materials and Methods

### *2.1. Location of the Study Area*

2.1.1. Experimental Area

The study was implemented in an experimental area of the Instituto Federal do Triângulo Mineiro (IFTM), campus Uberaba, Brazil, located at 19°39′43.4″ S and 47°57′57″ W, at 795 m above sea level, between September 2016 and August 2017.

In the past four years, the experimental area has been cultivated with cauliflower (Brassica oleracea var. botrytis), broccoli (*Brassica oleracea* var. *italica*), and cabbage (*Brassica oleracea* var. *capitata*) in the NTS, on the cultural residues of Brachiaria, Sunn hemp, and Pearl millet, which were always the same sown in the plots after each cycle. These vegetables were irrigated daily via conventional aspersion to keep the soil moisture close to field capacity.

2.1.2. Region Climate

The climate of the experimental region is classified as Aw-hot, humid summer, and cold, dry winter [18]. The annual average precipitation and temperature are 1600 mm and 22.6 °C, respectively. However, the yearly rainfall accumulated was 1571, and 1264 mm 2016, and 2017, respectively [19] (Figure 1).

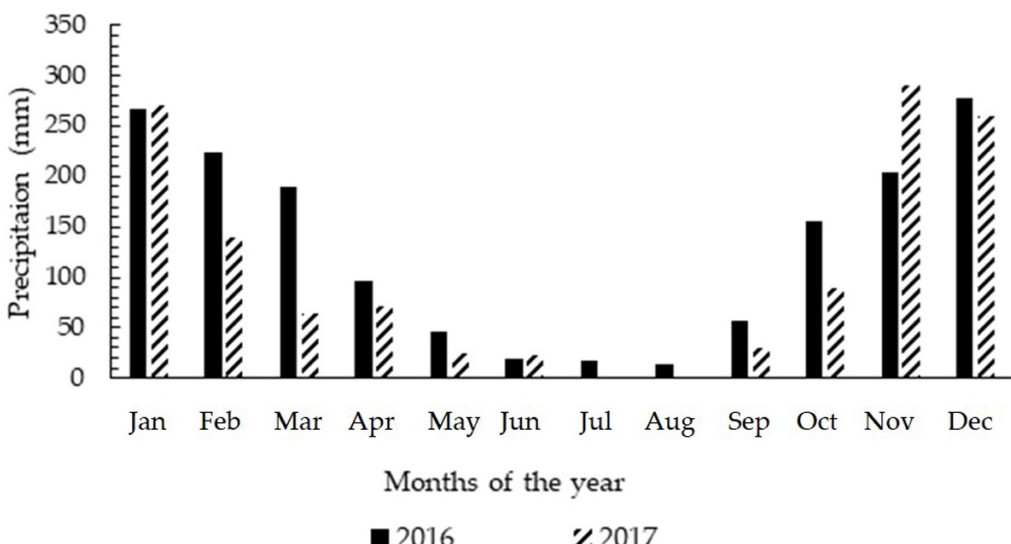

**Figure 1.** Monthly rainfall in 2016 and 2017 in Uberaba, MG. Brazil.

### 2.1.3. Soil Type

The soil of the experimental area was classified as a Oxisol [20], presenting in the 0–0.2 m soil layer: 210, 710, and 80 g kg$^{-1}$ of clay, sand, and silt, respectively. The soil chemical analysis indicated: pH CaCl$_2$ = 5.5; 76 mg dm$^{-3}$ of $p$ (resin); 2 mmolc dm$^{-3}$ of K$^+$; 22 mmolc dm$^{-3}$ of Ca$^{2+}$; 10 mmolc dm$^{-3}$ of Mg$^{2+}$; 17 mmolc dm$^{-3}$ of H$^+$ Al and 19 g dm$^{-3}$ of soil organic matter.

### 2.2. Experimental Design

The experimental design used was randomized blocks, six different coverages were evaluated: 1-brachiaria (B) (*Urochloa brizantha* cv Marandú), 2-Sunn hemp (S) (*Crotalaria juncea* L.), 3-Pearl millet (M) (*Pennisetum glaucum* L.), 4-S + B, 5-B + M, 6-S + M, with four replications. Each plot consisted of 60 m$^2$ (10 × 6 m$^2$), totalling 1080 m$^2$ of the experimental area. A Cerrado native area adjacent to the experimental study area was also soil sampled for comparative reasons.

### 2.3. Additional Information

The s seeding of the cover crops was mechanized using a Semina$^®$ 2 seeder, with five rows spaced 0.2 m apart. About 25, 50, and 50 seeds per meter of S, B, and M, respectively, were sown individually. The mixes (S + B, S + M, and B + M) were sown at 50% of the rate for individual cultivation.

### 2.4. Assessments

These cover crops were cultivated from September to November (Spring) without applying mineral or organic fertilizer. When 50% of the plants reached the maximum flowering, samplings were taken from a 2 m$^2$ area in each plot. The collected plant material was taken to an oven (forced air circulation), dried at 65 °C for 72 h, weighed, and the results were expressed in kg ha$^{-1}$. After cover crop sampling, the plants in each area were desiccated [2 kg ha$^{-1}$ of ammonium salt (792.5 g kg$^{-1}$ active ingredient as Roundup$^®$ WG) + 2 L ha$^{-1}$ of salt of dimethylamine of dichlorophenoxyacetic acid (840 g L$^{-1}$ active ingredient as 2.4-D Nortox$^®$)].

After the management of the coverings (drying), the coverings were cut close to the ground with a motorized costal cutter to lodge the straw, then holes were dug with a maximum depth of 18 cm, where the fertilizer incorporated into the soil was placed, to later make transplanting the seedlings.

### 2.4.1. Mechanical Resistance of Soil to Root Penetration

All soil sampling and evaluations were carried out at soil depths of 0–0.1, 0.1–0.2, 0.2–0.3, and 0.3–0.4 m before planting the vegetables. The soil mechanical resistance to penetration (RP) was determined using an impact penetrometer (model IAA-Planalsucar), with a load capacity of 150 kgf, an accuracy of 0.001 kgf, a rod length of 0.65 m, fitted with stainless steel cone (AISI 516) with an inclination of 30° and diameter of 0.2027 m [21]. The RP field data were obtained in numbers of impacts (dm$^{-1}$) converted to kgf cm$^{-2}$ using the formula: R (kgf cm$^{-2}$) = 5.6 + 6.98 N [22]; the values were then multiplied by 0.098 to convert to MPa [23].

### 2.4.2. Soil Density, Microporosity, Saturation Volume, Macroporosity and Total Porosity

The soil bulk density (SD) was determined in samples with undisturbed soil structure by the volumetric ring method, collected in rings of 48 mm in diameter by 53 mm height with the Uhland auger. The samples were weighed at field moisture, dried at 105 °C for 24 h, then weighed again. In the soil samples, the soil pore size distribution was determined. The samples were previously soaked for 24 h then subjected to the suction of a 0.6 m water column [24].

The microporosity (Mi) (m$^3$ m$^{-3}$) was obtained through samples submitted to water column tension, which removes the water from the macropores and after weighing, before and after going to the oven, obtaining the volume of macro and micropores contained in the sample (Equation (1)).

$$Mi = (a - b)/c \tag{1}$$

On what: a = Weight of the sample after being subjected to a tension of 100 hPa; b = dry sample weight (g); c = Cylinder volume (cm$^3$).

Through the difference between the saturation volume (VS) (Equation (2)) and the microporosity (Mi) (Equation (1)), the macroporosity (Ma) (m$^3$ m$^{-3}$) of the soil is obtained (Equation (3)) [24].

$$VS = (a - b)/c \tag{2}$$

$$Ma = VS - Mi \tag{3}$$

On what: a = Weight of the saturated soil sample (g); b = Dry sample weight (g); c = Cylinder volume (cm$^3$).

Total porosity (TP) (m$^3$ m$^{-3}$) corresponds to the total pore volume of the soil occupied by water and/or air and is calculated using the measurement of the weight of the soil sample after saturation and after the second drying in an oven. (Equation (4))

$$TP = Ma + Mi \tag{4}$$

### 2.4.3. Soil Moisture

Soil samples to evaluate water content were collected on the same day and soil depths. Two soil samples were collected per plot and homogenized to obtain wet and dry mass. The samples were weighed and placed for drying in a forced air circulation oven at 105 °C for 24 h. After getting the gravimetric moisture (Ug), this was multiplied by the soil bulk density to determine the soil volumetric moisture (VM) [24].

### 2.4.4. Stability of Aggregates and Their Indexes

The soil aggregates stability was evaluated by the methods described by [25], which comprises the weighing of soil samples of 50 g, air dried, in duplicates (soil aggregates with diameter between 4.76 to 9.51 mm), which were moistened by capillarity during 12 h. The resulting soil material was transferred to a set of sieves of 4, 2, 1, 0.5, 0.25, and 0.13 mm mesh and subjected to vertical oscillation for 15 min in water. Each sieve's contents were separated into aluminum cans, dried in a forced-air circulation oven at 105 °C for 24 h, and then weighed.

The mean weight diameter (MWD) (Equation (5)) was calculated with the masses of the soil aggregates, which is great when a great percentage of large aggregates are retained in the sieves with larger meshes.

$$MWD = \Sigma(xi \times wi) \tag{5}$$

where, MWD = mean weight diameter; xi = average diameter of classes (mm); wi = proportion of each aggregate class.

The geometric mean diameter (GMD) (Equation (6)) estimates the class size of aggregates of superior occurrence.

$$GMD = exp\{\Sigma[(ln[xi][pi])]/\Sigma[pi]\} \tag{6}$$

where, ln[xi] = natural logarithm of average diameter of classes; pi = weight retained on each sieve (g); xi = average diameter of the classe (mm).

The aggregate stability index (ASI) (Equation (7)) represents the proportion of soil aggregates larger than 2 mm [26].

$$ASI = [(SW - WA)/(SW)] \times 100 \tag{7}$$

where, SW = sample weight (g); WA = weight of aggregates <0.25 mm (g).

### 2.5. Statistical Analysis

The principal components analysis (PCA) using SD, Ma, Mi, TP, MWD, GMD, and ASI attributes were analyzed, followed by an R program cluster analysis. The analysis of variance (F test, $p < 0.05$) was performed for the factorial cover crops and soil depth using the Sisvar program; when significant differences were identified, the averages were compared by the Scott-Knott test ($p < 0.05$).

## 3. Results

### 3.1. The Soil Compaction Indicators

The mechanical resistance to soil penetration presented significant differences ($p < 0.05$) among treatments; however, no compacted soil layer was observed up to 0.4 m soil depth. While the observed SD value ranged from 1.61 to 1.78 kg dm$^{-3}$, except for M at depths of 0–0.1 and 0.3–0.4 m (Table 1).

The lowest values of RP and SD were observed in treatments with cover crops of the Poaceae family (B and M) when cultivated alone (single) compared to single Sunn hemp (Fabaceae), for depths of 0.2–0.3 and 0.3–0.4 m (Table 1). In addition, this result may be due to the effect of the Poaceae root system, which has voluminous and well-developed roots.

The volumetric moisture content (VM) was constant for all soil depths and close to field capacity, and there were no significant differences between the evaluated covers and depths.

**Table 1.** Soil resistance to penetration (RP), soil bulk density (SD) and volumetric moisture (VM) in a Oxisol cultivated with different cover crops.

| Cover Crops | RP | SD | VM |
|---|---|---|---|
| | Mpa | kg dm$^{-3}$ | cm$^3$ cm$^{-3}$ |
| | | 0–0.1 m | |
| Brachiaria (B) | 1.94 aA | 1.68 bB | 0.25 aA |
| Sunn hemp (S) | 2.17 aA | 1.74 bA | 0.26 aA |
| Pearl millet (M) | 1.66 aB | 1.43 aC | 0.27 aA |
| B + S | 1.43 aC | 1.72 bB | 0.26 aA |
| B + M | 1.62 aA | 1.67 bA | 0.26 aA |
| M + S | 1.89 aA | 1.76 bA | 0.28 aA |

**Table 1.** *Cont.*

| Cover Crops | RP | SD | VM |
|---|---|---|---|
| | **Mpa** | **kg dm$^{-3}$** | **cm$^3$ cm$^{-3}$** |
| | | 0.1–0.2 m | |
| Brachiaria (B) | 1.66 aB | 1.78 aA | 0.25 aA |
| Sunn hemp (S) | 1.80 aB | 1.71 bA | 0.29 aA |
| Pearl millet (M) | 1.89 aA | 1.73 bA | 0.28 aA |
| B + S | 2.17 aA | 1.77 aA | 0.27 aA |
| B + M | 1.61 aA | 1.71 bA | 0.26 aA |
| M + S | 1.75 aB | 1.75 bA | 0.27 aA |
| | | 0.2–0.3 m | |
| Brachiaria (B) | 1.39 aC | 1.73 bA | 0.28 aA |
| Sunn hemp (S) | 1.71 bC | 1.77 aA | 0.30 aA |
| Pearl millet (M) | 1.57 aB | 1.69 bA | 0.27 aA |
| B + S | 1.89 bA | 1.70 bB | 0.28 aA |
| B + M | 1.48 aB | 1.73 bA | 0.28 aA |
| M + S | 1.52 aC | 1.66 bA | 0.27 aA |
| | | 0.3–0.4 m | |
| Brachiaria (B) | 1.34 aC | 1.61 aB | 0.27 aA |
| Sunn hemp (S) | 1.71 bC | 1.65 bA | 0.30 aA |
| Pearl millet (M) | 1.38 aC | 1.53 aB | 0.26 aA |
| B + S | 1.66 bB | 1.71 bB | 0.27 aA |
| B + M | 1.25 aC | 1.70 bA | 0.29 aA |
| M + S | 1.43 aD | 1.58 aB | 0.28 aA |
| CV (%) | 4.24 | 5.60 | 14.10 |

Averages followed by the same letter, lowercase comparing cover crops and uppercase comparing soil layers, do not differ among each other by the test of Scott-Knott ($p < 0.05$). CV = coefficient of variation.

## 3.2. The Correlations between Physical Soil Attributes

The variables RP, SD, and VM significantly correlated with each other, positive for RP and SD and negative for RP and MV in depths up to 0.3 m, and positive for SD and VM up to 0.40 m depth (Table 2). When Pearson's correlation is positive, as it occurs with RP × SD, and SD × VM, the variables are interconnected, increasing or decreasing simultaneously.

**Table 2.** Pearson's correlations between the values of soil resistance to penetration (RP), soil bulk density (SD), and volumetric moisture (VM) between treatments.

| Variables | Pearson Correlation | Significance |
|---|---|---|
| | **r$^2$** | **%** |
| | 0–0.1 m | |
| RP × SD | 0.31 ns | ns |
| RP × VM | −0.54 ** | 99 ** |
| SD × VM | 0.91 ** | 99 ** |
| | 0.1–0.2 m | |
| RP × SD | 0.32 * | 99 ** |
| RP × VM | −0.37 ** | 99 ** |
| SD × VM | 0.38 ns | ns |
| | 0.2–0.3 m | |
| RP × SD | 0.12 * | 99 ** |
| RP × VM | −0.21 ** | 99 ** |
| SD × VM | 0.03 ns | ns |
| | 0.3–0.4 m | |
| RP × SD | 0.30 ns | ns |
| RP × VM | 0.12 ** | 99 ** |
| SD × VM | 0.24 ns | ns |

* = significant at 5%; ** = significant at 1%; ns = non-significant.

The physical attributes and the indicators of soil aggregation presented significant interactions, such as Ma × SD, TP × SD, TP × Ma, MWD × SD, GMD × ASI, and MWD × GMD, at 0–0.1 m soil depth (Table 3). These results indicate that the soil quality can be defined by different attributes that are correlated.

**Table 3.** Pearson's correlations between soil physical attributes under different cover crops and Cerrado native area (0–0.4 m soil depth).

| Variables | Ma | Mi | TP | MWD | GMD | ASI |
|---|---|---|---|---|---|---|
| | | | 0–0.1 m | | | |
| SD | −0.86 * | −0.11 | −0.97 * | −0.74 | −0.77 * | −0.68 |
| Ma | | −0.35 | 0.85 * | 0.44 | 0.49 | 0.50 |
| Mi | | | 0.20 | 0.31 | 0.28 | 0.20 |
| TP | | | | 0.63 | 0.68 | 0.61 |
| MWD | | | | | 0.99 * | 0.74 |
| GMD | | | | | | 0.74 |
| | | | 0.1–0.2 m | | | |
| SD | −1 * | −0.70 | −0.97 * | −0.68 | −0.27 | 0.96 * |
| Ma | | 0.68 | 0.97 * | 0.63 | 0.21 | −0.97 * |
| Mi | | | 0.84 * | 0.22 | −0.19 | −0.81 * |
| TP | | | | 0.54 | 0.09 | −0.99 * |
| MWD | | | | | 0.89 * | −0.47 |
| GMD | | | | | | 0 |
| | | | 0.2–0.3 m | | | |
| SD | −0.95 * | −0.97 * | −0.98 * | −0.14 | 0.06 | 0.84 * |
| Ma | | 0.93 * | 0.97 * | −0.12 | −0.31 | −0.93 * |
| Mi | | | 0.99 * | −0.01 | −0.21 | −0.90 * |
| TP | | | | −0.06 | −0.25 | −0.93 * |
| MWD | | | | | 0.98* | 0.43 |
| GMD | | | | | | 0.60 |
| | | | 0.3–0.4 m | | | |
| SD | −0.85 * | −0.69 | −0.96 * | 0.25 | 0.33 | 0.52 |
| Ma | | 0.26 | 0.94 * | −0.71 | −0.77 * | −0.87 * |
| Mi | | | 0.58 | 0.35 | 0.29 | 0.06 |
| TP | | | | −0.47 | −0.55 | −0.72 |
| MWD | | | | | 1 * | 0.95 * |
| GMD | | | | | | 0.97 * |

* = significant at 5%. SD = soil bulk density; VM = soil volumetric moisture; Ma = soil macroporosity; Mi = soil microporosity; TP = soil total porosity; MWD = mean weight diameter; GMD = geometric mean diameter; ASI = aggregate stability index.

The results also showed a positive correlation between ASI and GMD, and between MWD and GMD, demonstrating that the increase of soil aggregate stability was related to the increase of the GMD, consequently, the increase of MWD.

The evaluation of the Ma, TP, GMD, and MWD in the 0.1–0.2 m soil layer demonstrated a negative correlation with SD (Table 3), since the higher the value for these indicators, the less will be the SD. These negative correlations illustrate what happens in reality; for example, the greater amount of soil macropores, the lower the SD will be.

### 3.3. Analysis of the Main Components

The principal components analysis and grouping tests enable the distribution of these physical attributes and indicators of aggregation at 0–0.1 m soil depth (Figure 2), distinguishing the cover crops in different groups, taking the Cerrado native area as a basis of comparison as an ideal environment. This Cerrado native area presents the characteristics that exemplify a preserved soil, with a supposed ideal soil physical quality condition for root growth and plant development.

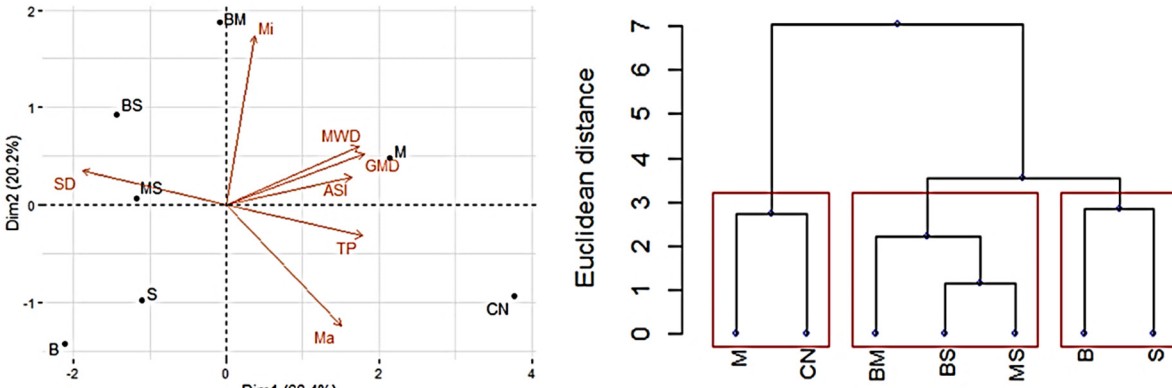

**Figure 2.** Principal components analysis, and cluster analysis of cover crops and Cerrado native area (CN) (0–0.1 m soil depth). B = Brachiaria, M = Pearl millet, S = Sunn hemp; BM = B + M; BS = B + S; MS = M + S; CN = Cerrado native forest. SD = soil bulk density; Ma = soil macroporosity; Mi = soil microporosity; TP = soil total porosity; MWD = mean weight diameter; GMD = geometric mean diameter; ASI = aggregate stability index.

There was a significant improvement in soil aggregation in the area covered with M, presenting results similar to those found in the Cerrado native forest, which presented ASI of 0.959 and 0.952; GMD of 1.70 and 1.77 mm, and MWD of 4.31 and 4.58 mm, respectively.

The formation of groups of treatments (cluster analysis) of the principal components analysis indicated that none of the cover crops was similar to the Cerrado native area (Figure 3), all grouped in a single distinct group. The soil physical quality found in the 0.1–0.2 m layer under the cover crops evaluated was not similar to the results found in the Cerrado native area. However, the formation of two distinct groups stands out, being with the Poaceae only (B, M, and B + M) and with the single Fabaceae (S) or consortium (S + M, B + S).

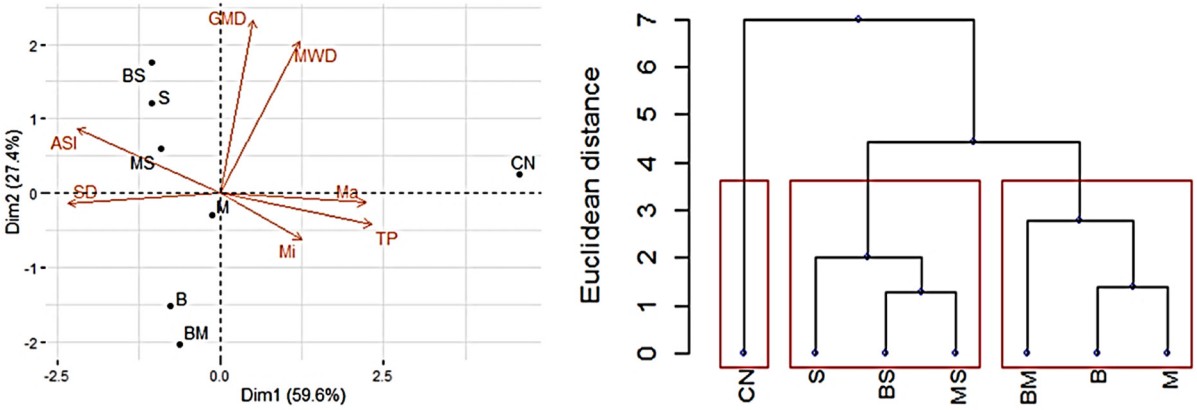

**Figure 3.** Principal components analysis, and cluster analysis of cover crops and Cerrado native area (CN) (0.1–0.2 m soil depth). M = pearl millet, B = brachiaria, S = sunn hemp; BM = B + M; BS = B + S; MS = M + S; CN = Cerrado native. SD = Soil bulk density; Ma = Soil macroporosity; Mi = Soil microporosity; TP = Soil total porosity; MWD = Mean weight diameter; GMD = Geometric mean diameter; ASI = Aggregate stability index.

The SD for B = 1.78; M = 1.73; S = 1.72; B + M = 1.71; B + S = 1.77, and M + S = 1.75 kg dm$^{-3}$, was considerably higher than the SD found in the Cerrado native area (1.38 kg dm$^{-3}$). This fact is also evidenced by the values of Ma and TP on B = 11.91 and 48.65%; M = 12.58 and 49.64%; Sunn hemp = 14.01 and 44.02%; B + M = 15.39 and 47.73%; B + S = 12.25 and 45.76%, and M + S = 11.33 and 46.14%, when compared to the Cerrado native area of 22.54 and 59.81%, respectively.

In the 0.2–0.3 m soil layer, similar to what occurred in the 0.1–0.2 m soil layer, no cover crop treatment has reached the characteristics obtained in the Cerrado native forest (Figure 4). This can be justified by the short implementation time of the NTS in the area (4 years), which is necessary to observe the evolution in soil physical quality in this soil layer. However, at this depth, it is possible to observe the formation of a group composed only of Poaceae species (B, M, and B + M), differing from the other plant species.

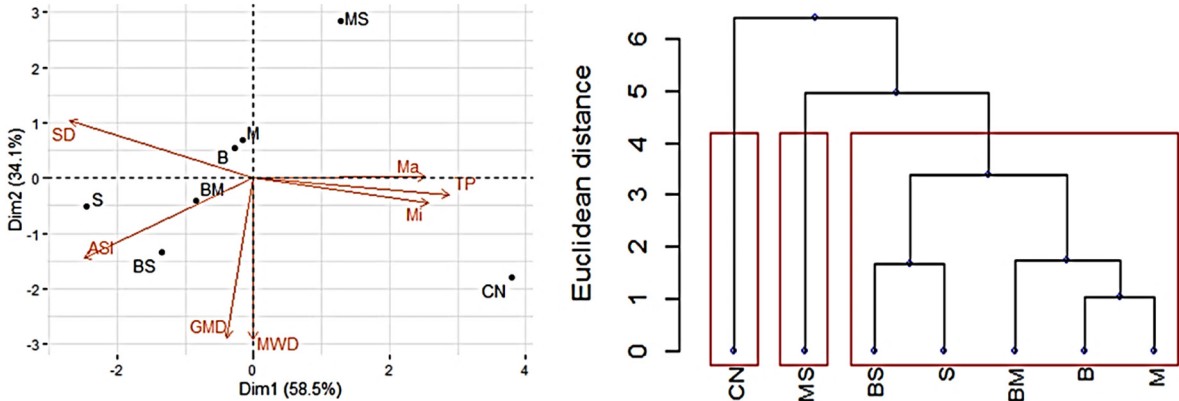

**Figure 4.** Principal components analysis, and cluster analysis of cover crops and Cerrado native area (CN) (0.2–0.3 m soil depth). B = Brachiaria, M = Pearl millet, S = Sunn hemp; BM = B + M; BS = B + S; MS = M + S; CN = Cerrado native. SD = soil bulk density; Ma = soil macroporosity; Mi = Soil microporosity; TP = Soil total porosity; MWD = Mean weight diameter; GMD = Geometric mean diameter; ASI = Aggregate stability index.

The analysis of the principal components and subsequent cluster analysis indicated the cover crop mix 'B + M' as the most favorable conditions among the physical attributes, for the soil in the layer of 0.3–0.4 m, and even above the results found in the Cerrado native area (Figure 5).

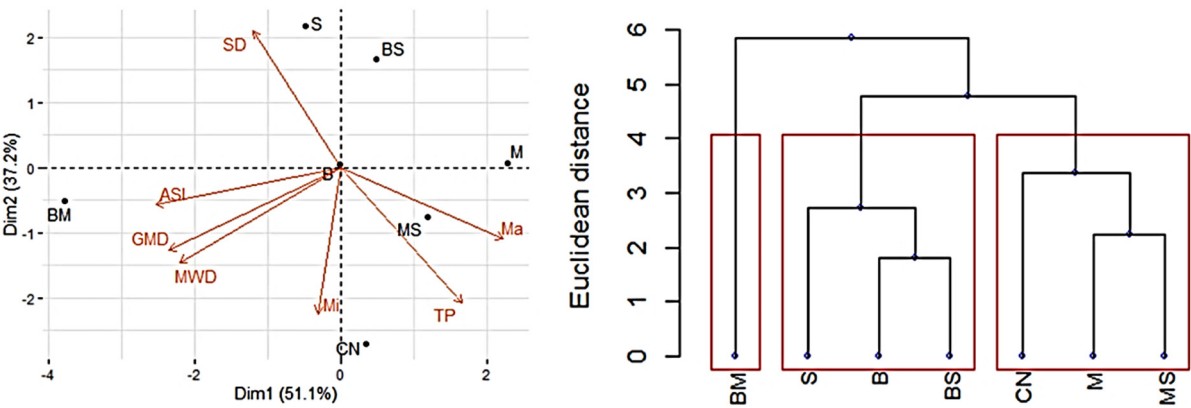

**Figure 5.** Principal components analysis and cluster analysis of cover crops and Cerrado native area (CN) (0.3–0.4 m soil depth). B = Brachiaria, M = Pearl millet, S = Sunn hemp; BM = B + M; BS = B + S; MS = M + S; CN = Cerrado native. SD = soil bulk density; Ma = soil macroporosity; Mi = Soil microporosity; TP = Soil total porosity; MWD = Mean weight diameter; GMD = Geometric mean diameter; ASI = Aggregate stability index.

The soil aggregation indicators demonstrate the superiority of the cover crop mixtures in relation to the isolated crop with only one species, since the best results for ASI (0.94), GMD (1.63 mm) and MWD (4.05 mm) were observed in the 'B + M' mixture when compared to B = 0.89, 1.30 and 3.20; M = 0.87, 1.14 and 2.61; Sunn hemp = 0.91, 1.27 and 3.03;

B + S = 0.88, 1.22 and 2.92; M + S = 0.89, 1.31 and 3.26; Cerrado native = 0.91, 1.43 and 3.61 for ASI, GMD (mm) and MWD (mm), respectively.

## 4. Discussion

### 4.1. The Physical Attributes of the Soil and Its Indicators and Quality

The existing similarity of RP among soil layers can be justified by the short period that this area has been cultivated under the NTS of soil management; also, the root systems of the cover crops provide positive effects to the soil. Root systems perform a biological soil decompression over the years of cultivation, similarly to the results found by [3,15].

Generally, the RP values of around 2.5 Mpa are considered low, however, values around 3.5 to 6.5 Mpa are considered damaging to Fabaceae and Poaceae plant species [27]. In this study, the RP for all cover crops evaluated was below these levels considered critical for root development, indicating that even after four cultivation cycles of mechanically sown cover crops and manual planting of brassicas, there was no significant increase in RP, with this it can be inferred that the root system of these plants is acting in the restructuring of the soil, increasing the aggregate stability, and minimizing compaction problems. (Table 1).

The evaluation of forms of soil mechanical decompression with ripper and subsoiler associated with biological decompression with M and H indicated that any of the treatments were able to influence any of the maize agronomic characteristics [28]. The author also observed that even in areas with soil RP up to 3.3 Mpa, no form of decompression was necessary since maize characteristics and productivity were not affected by such RP.

The correlation between soil compaction and the time of NTS installation was studied by [29], who found that in areas with NTS installed for five and 12 years, the soil layers of 0.1–0.2 and 0.2–0.3 m presented soil compaction critical level. Compared to those obtained in the present study, these divergent results can be explained by the low machinery traffic in the area for the production of vegetables.

The roots of Poaceae, when decomposing, can form biopores and, thus, improve soil aggregation, with an increase in macroporosity and a decrease in SD and RP [30]. These authors found that the ground cover plants of the Poaceae family promoted a greater volume of macropores when compared to the plants of the Fabaceae family.

The soil physical quality of an Oxisol was similar between no-tillage and conventional tillage systems in the semiarid [31], however, the authors found that the higher compression rate occurred in the superficial soil layers under NTS. Still, the RP observed (<2 Mpa) was insufficient to impair the plant species' root development evaluated. Many authors consider this value (<2 Mpa) as the RP limit from which starts to occur restrictions to plant growth [7,23]; however, these effects are more harmful when the soil is in low humidity [32].

In irrigated areas, it is common for soil moisture to present similar values in all areas evaluated, as occurred in this study, which were between field capacity and the permanent wilting point, which is the ideal condition to make better use of the impact penetrometer's potential, since that in very humid soils there is no differentiation of measurement and in very dry soils the measurement is very time consuming and hampered by excessive soil resistance [33].

All SD values were above 1.60 kg dm$^{-3}$, this value is indicated by [34] as critical for developing these cultures. Ref. [35] proposed critical values for SD, characterizing the soil as compressed when SD reaches 1.55 kg dm$^{-3}$ for soil clay content ranging between 200 and 550 g kg$^{-1}$, within the range of the clay content found in the soil of this study.

The SD in areas under different vegetation, among them pearl millet, brachiaria and Sunn hemp demonstrated that no differences occurred among treatments (cover crops) in 0–0.1 and 0.1–0.2 m soil layers of an Oxisol [36]. Ref. [14] showed that the highest rates of SD and RP occurred at a depth of 0.08–0.15 m, concluding that the increase in SD interfered with root growth, reduced aeration, caused a rise in RP, and caused changes in the dynamics of water in the soil.

However, Refs. [1,3] observed that SD of an Oxisol cultivated with M and H decreased in the 0.1–0.2 m soil layer. In a similar study, Ref. [37] also found no differences among

the treatments (fallow, Sunn hemp, pigeon pea, *Mucuna* sp.; sorghum) for SD in layers of 0–0.1 and 0.1–0.2 m, similarly to the SD found in the present study. It should be noted that M (0–0.1 m and 0.3–0.4 m) was the only treatment that presented SD values lower than 1.55 kg dm$^{-3}$, as highlighted by [35] as critical values for SD. This indicates that M can be used as a cover plant in the Cerrado area to reduce SD values.

In the present study, the RP was 2.17 MPa in sunn hemp (0–0.1 m) and in 'B + S' (0.1–0.2 m), and the SD were 1.74 and 1.77 kg dm$^{-3}$, respectively (Table 2). A negative Pearson's correlation indicates that the other decreases as one increase. In the present study, the RP negatively correlates with the VM up to 0.3 m soil depth, where the RP increased the VM decreases, indicating that water retention is low in this soil layer.

The soil physical quality in commercial sugarcane areas presented similar results regarding the positive correlation between the SD and RP, both at 5° and 6° cutting sugar cane [38]. However, in the same study, the correlation between RP and VM, and VM and SD were negative, contrasting with the results observed in the present study at 0–0.1 and 0.3–0.4 m soil depth, and resembling those observed at 0.1–0.2 and 0.2–0.3 m for the negative interaction between RP and VM. Ref. [32] highlight that there will be a greater RP and SD in a smaller VM.

The soil attributes that negatively correlate at 0–0.1 m soil depths were SD × Ma and TP × MWD. These correlations indicated that an increase of SD will cause a decrease in soil macroporosity and that a rise in soil TP will cause a reduction in MWD (Table 3), which shows that the soil macroporosity is directly related to a decrease in the size of the soil aggregates, as also observed by [7].

The positive correlation between TP and Ma demonstrates that crop managements affect soil macroporosity and the total volume of soil pores (Table 3). Ref. [11] found similar results, where the reduction of SD occurred simultaneously during sugarcane seasons, and the total soil porosity increased.

The evaluation of the Ma, TP, GMD and MWD in the 0.1–0.2 m soil layer demonstrated a negative correlation with SD (Table 3), i.e., the higher the value for these indicators the less will be the SD, exemplifying what happens in reality, for example, as for the greater amount of soil macropores the lower will be the SD.

The soil attributes ASI and GMD, and between MWD and GMD, indicate the conditions of the soil particle aggregation and correlate directly to the extent that if only one attribute is improved, the others are also benefited.

Again, in the 0.2–0.3 m soil layer, positive and negative correlations were observed among the variables analyzed. Positive correlations occurred between attributes Mi × Ma, TP × Ma, and TP × Mi, highlighting that all soil attributes correlated with the soil porosity are incremented or reduced in conjunction; the same is seen for the indicators soil aggregation MWD and GMD (Table 3).

In contrast, the Ma × SD, Mi × SD, were negatively correlated, pointing out that soil particles' approximation will reduce soil pore abundance. The ASI also showed a negative correlation with Ma, Mi, and TP, indicating that the soil porosity generally reduces with soil aggregation and vice-versa. The indicators can prove the loss of soil quality because the decrease in TP, SD proves particle densification. Even increasing the ASI, there may be a negative change in the dynamics of air and water in the soil. This situation difficulties the plant root development, as observed by [2,3,6].

In the 0.3–0.4 m, the soil attributes that presented positive correlation were TP × Ma, GMD × ASI, MWD × ASI, MWD × GMD, and the negative correlations were Ma × SD, Mi × SD, TP × SD, ASI × Ma, and GMD × Ma (Table 3). Such correlations show that the increase of SD directly contributes to the reduction of Ma, Mi, and TP, i.e., a decrease in soil porosity is directly linked to the size of the aggregates. It is also possible to check that the indicators of soil aggregation (GMD, MWD, and ASI) are positively correlated among each other, i.e., the gradual increase of one attribute will cause the rise of the other indicator [3,5,6].

### 4.2. The Principal Components Analysis and Their Interpretations

Among cover crops, millet and native Cerrado forest differ from other treatments in a different group (Figure 2), which shows the improvement provided to the system by the millet (fasciculated) root system, which is deep and broad, directly influencing the decompaction of the soil and consequently, reducing the DS, these results are similar to those obtained by [2,3], where they observed that the occurrence of improvements in all evaluated attributes.

In their study, Ref. [39] observed that in areas cultivated with Poaceae plants (pasture), there were higher proportions of stable aggregates in water in the 8 to 2 mm class compared to native forest. This observation was attributed to the influence of the fasciculated root system, which irradiates in the soil profile and releases large amounts of exudates in the soil. These exudates improve the connections between the mineral particles and the soil aggregate formation.

Comparing NTS, crop-livestock integration (ILP), and native forest (NF), Ref. [40] observed that, regardless of depth, the NF area had the highest values of MWD, GMD, and ASI. This pattern was due to the great supply of plant material in the area under original soil conditions, without any cultivation interference.

The similarity between the Peal millet area and the Cerrado native area is probably due to the root system of the Poaceae plant species that is more efficient in increasing and maintaining the stability of soil aggregates compared to Fabaceae [9]. This contribution of the Poaceae can be evidenced in the grouping analysis, where M is grouped with Cerrado (CN). Among the intercropped cover crops, only the Poaceae plant species (B + M) are separated from the mixtures with Fabaceae (B + S, M+ S).

Such a positive contribution to soil aggregation facilitates crop development and root penetration, thus favoring the expansion and diversification of the soil microbiota [41]. In studies evaluating soybean preceded by the cultivation of different cover crops, Ref. [34] found that M cultivation favored the soybean root growth below compacted soil layers.

The results observed in the formation of treatment groups (cluster analysis) indicate that the plants of the Poaceae family are more efficient in improving the physical attributes of the soil (Figure 3), as highlighted by [9,30].

The short time after introducing the NTS, succeeding conventional tillage system (CTS), justifies the difference in soil quality between the physical soil layers of 0–0.1 and 0.1–0.2 m. Besides, the soil under the Cerrado native area provides the best conditions for SD and TP since this area does not suffer from heavy machinery traffic found in the no-tillage system.

The influence of different soil management systems and the use of cover crops on the organic production of beans and maize presented similar results to those obtained in the present study, according to [37]. The authors observed no difference between the crop covers for SD, Ma, and Mi, but observed lower SD and higher Ma and TP in a native forest than under NTS in the 0.1–0.2 m soil layer.

According to [42,43], in their studies comparing areas in CTS and NTS, observed that the activities developed with agricultural machines and implements caused an adverse change in the soil structure, reducing the TP and the size of the soil particles, consequently reducing Ma and increasing Mi.

The promising results observed when cultivated mixed cover crops, such as B + M, are probably due to the joint action of B and M's root systems. Two Poaceae plant species in the same area amplified the effects of the cover crops on the soil aggregation in deep soil layers (0.3–0.4 m).

In their study, Ref. [41] studied the effects of different cover crops on the soil's physical attributes. They concluded that plants with aggressive root systems (Poaceae plant species) could improve soil physical attributes. These soil improvements corroborate the present study results, where M (Poaceae) presented the best results for soil attribute improvement. This result can also be seen in the cluster analysis, as there was a grouping with CN, M, and

M + S, indicating that the presence of M favored the improvement of physical attributes similar to the native area (CN).

## 5. Conclusions

Low resistance to penetration and soil density were observed for Brachiaria and Pearl millet (Poaceae) compared to Sunn hemp (Fabaceae) in deeper soil layers;

The principal components and cluster analysis indicated Brachiaria + Pearl millet as the most promising coverage for deep soil structuring.

The soil physical quality indicators showed that millet in isolated cultivation or intercropped with another cover was the culture that presented the best results for most of the evaluated characteristics.

The best indices of soil aggregation were observed where the species of the Poaceae Family were being cultivated in isolation or intercropped with each other.

**Author Contributions:** Conceptualization, J.L.R.T., J.C.M.J.; Methodology and Experiment Implementation, B.d.S.S., G.L.d.S. and D.M.d.S.V.; Formal Analysis, J.L.R.T., E.M.L.; Data Curation, J.L.R.T., E.M.L.; Writing-Original Draft, J.L.R.T., A.L.; Writing-Review and Editing, E.M.L.; Project Administration, J.L.R.T., R.L.d.A.; Funding Acquisition, J.L.R.T. All authors have read and agreed to the published version of the manuscript.

**Funding:** This research was funded by FUNDAÇÃO AGRISUS, project PA 2201/17, for a study period of 24 months, starting on 10 January 2017, with the signature of the Grant Term by its legal representative.

**Institutional Review Board Statement:** Not applicable.

**Informed Consent Statement:** Not applicable.

**Data Availability Statement:** Data is contained within the article.

**Acknowledgments:** The authors are grateful to the Federal Institute of Education, Science and Technology of the Mineiro Triangle for providing the necessary equipment and laboratory space to conduct the experiments and analyses; to the Luiz de Queiroz Foundation for Agricultural Studies (AGRISUS FOUNDATION); the Research Support Foundation of Minas Gerais State (FAPEMIG)and National Council for Scientific and Technological Development (CNPq), for granting scholarships and Scientific Initiation to the students involved with the project.

**Conflicts of Interest:** The authors declare no conflict of interest regarding the data and findings published in this article.

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
