# Peer review of "Physical Attributes of an Irrigated Oxisol after Brassicas Crops under No-Tillage System"

_agronomy, doi:10.3390/agronomy12081825_

Round 1
Reviewer 1 Report
The author tries to analyze the effect of cover crops on the physical attributes, which is valuable in conservation tillage areas. And there are some problems as following:
1. From the title and Introduction, irrigated area is an important factor for the paper, but it is not considered in the experiment design and results. What does the author think about this?
2. In the third and fourth paragraphs of the Introduction, Fabaceae and Poaceae plant species are proposed from the perspective of compaction. However, as two commonly used green manures, Fabaceae and Poaceae plant species not only play a role in alleviating compaction. It is suggested that the author rearrange the contents of this part and explain clearly why Fabaceae and Poaceae plant species are selected.
3. In section 2.1.1, the experimental period is in 2016 and 2017, while in Section 2.1.2, it is written in 2014-2017. What is the actual experimental period? If the experiment is only for two years and the growing season is only two months (September to November) each year, it is not suitable to use successive cultivation in title.
4. During the experimental period, are any other crops grown except the cover crops? Whether it will affect the change of soil physical properties?
5. what is the amount of each cover crop in section 2.2?
6. It is suggested that the pictures of these six cover crops are added in the paper, so that readers can have a more intuitive understanding.
7. In Experiment design, six coverages with four replications, that means 24 plots totally. Each plot consisted of 60 m2, so the experimental area should be 1440, which is inconsistent with the 1080 mentioned in the text. Please check.
8. The expression of soli depth in Table 1 is easy to confusing, and suggest a better form of expression.
9. The root system of cover crops has a great influence on soil physical properties, so whether the paper has sampled and analyzed the root system?
10. P288 “values around 3.5 to 6.5 Mpa are considered damaging to Fabaceae and Poaceae plant species(TORRES and SARAIVA, 1999). In this study, the RP for all cover crops evaluated was below these levels considered critical for root development, indicating the beneficial contribution of the NTS for soil improvement and conservation (Table 1).” Causality is unclear.
11. Lines 293-303 seems to have no relationship with the paper, please reorganize or delete.
Author Response
REPLIES TO REVIEWER 1.
- From the title and Introduction, irrigated area is an important factor for the paper, but it is not considered in the experiment design and results. What does the author think about this?
The irrigated area is a very important factor in the study, as it changes soil moisture and temperature, with this, the processes that involve decomposition, nutrient cycling and processes that involve organic matter occur more quickly, because in these areas the quantity and biodiversity of organisms in the soil increases considerably, with this, the changes that occur in the physical and chemical attributes of the soil are also affected.
- In the third and fourth paragraphs of the Introduction, Fabaceae and Poaceae plant species are proposed from the perspective of compaction. However, as two commonly used green manures, Fabaceae and Poaceae plant species not only play a role in alleviating compaction. It is suggested that the author rearrange the contents of this part and explain clearly why Fabaceae and Poaceae plant species are selected.
The reviewer was indeed right in his comments and the text has been redrafted, highlighting the other important roles that these plants play in these areas.
- In section 2.1.1, the experimental period is in 2016 and 2017, while in Section 2.1.2, it is written in 2014-2017. What is the actual experimental period? If the experiment is only for two years and the growing season is only two months (September to November) each year, it is not suitable to use successive cultivation in title.
Title, paragraph and figure were adjusted according to the reviewer's suggestion, as the study was conducted only in the years 2016 and 2017.
- During the experimental period, are any other crops grown except the cover crops? Whether it will affect the change of soil physical properties?
This issue was better clarified in the text, as the only crops that were mechanically cultivated and always in the same area were the coverings, while the brassicas were planted in holes, with most of their root system being removed at the time of harvest.
- what is the amount of each cover crop in section 2.2?
The number of plants sown per meter was highlighted in the text.
- It is suggested that the pictures of these six cover crops are added in the paper, so that readers can have a more intuitive understanding.
No photographs were found that contemplated all coverage and mixtures.
- In Experiment design, six coverages with four replications, that means 24 plots totally. Each plot consisted of 60 m2, so the experimental area should be 1440, which is inconsistent with the 1080 mentioned in the text. Please check.
The text has been properly corrected.
- The expression of soli depth in Table 1 is easy to confusing, and suggest a better form of expression.
The text has been properly corrected.
- The root system of cover crops has a great influence on soil physical properties, so whether the paper has sampled and analyzed the root system?
The root system was not sampled or evaluated in this study.
- P288 “values around 3.5 to 6.5 Mpa are considered damaging to Fabaceae and Poaceae plant species(TORRES and SARAIVA, 1999). In this study, the RP for all cover crops evaluated was below these levels considered critical for root development, indicating the beneficial contribution of the NTS for soil improvement and conservation (Table 1).” Causality is unclear.
The text was redone and the issue was clarified and correlated.
- Lines 293-303 seems to have no relationship with the paper, please reorganize or delete.
The text has been reorganized.

Reviewer 2 Report
No-tillage system is crucial for crop production and soil sustainability. Litter return to soil improve soil physical quality. This paper evaluated the changes of soil physical properties under six cover crops. The results are impressive and I think this paper is high quality. Also the writing is excellent.
Here are several corrections need to make.
(1) Methods of microporosity measurement should be described with details in section 2.4.2.
(2) Soil moisture is in highly dynamics. It is also high related with soil pore system, particular with soil total porosity. Measurement of soil moisture can only show the soil water status at sampling time. In this paper, soil moisture was measured once. I do not think this data is informative. I suggest this data can omit or if you have more data of soil moisture at varied vegetation grow stages, it is better to add more data.
Author Response
REPLIES TO REVIEWER 2
(1) Methods of microporosity measurement should be described with details in section 2.4.2.
The formulas and forms that were used to evaluate the physical attributes were better detailed in the material and methods.
(2) Soil moisture is in highly dynamics. It is also high related with soil pore system, particular with soil total porosity. Measurement of soil moisture can only show the soil water status at sampling time. In this paper, soil moisture was measured once. I do not think this data is informative. I suggest this data can omit or if you have more data of soil moisture at varied vegetation grow stages, it is better to add more data.
The determination of moisture at the time of the evaluation of this study is fundamental, as it directly affects the evaluation of soil compaction indicators, especially penetration resistance, it also affects soil temperature and moisture, with this, the processes that involve the organic matter happen more quickly and consequently there will be important changes in the attributes of the soil as well.
Yours sincerely
______________________
José Luiz Rodrigues Torres
corresponding author

Reviewer 3 Report
Comments in the attachment

Author Response
REPLIES TO REVIEWER 3
- The condition of the physical properties of agricultural soils is influenced by agricultural machinery. Have you considered this aspect?
The 2nd, 3rd and 4th paragraphs of the literature review deal with the importance and influence of agricultural machines on soil compaction in areas with and without irrigation, presenting the issue and its problems to the reader.
The methodology explains how soil compaction was estimated, behind the index that indicates the layer where the mechanical resistance to penetration of the roots into the soil occurs, performed with the impact penetrometer (Stolf et al., 2014). In the results, the occurrence of the compaction problem was presented, correlated with other physical attributes and discussed according to the international literature.
- How does irrigation affect agrophysical properties?
The irrigation indirectly affects the physical, chemical and biological attributes of the soil, as it changes the humidity and consequently the temperature, providing adequate conditions for the increase in quantity and diversity of soil organisms.
The decomposition/mineralization of organic matter is directly related to temperature and soil moisture. According to Lal et al. (1998, 2005 and 2006) the processes that involve organic matter in soils in tropical regions are 10 times faster when compared to those in temperate climates, due to adequate temperature and humidity for most of the year.
In irrigated areas, the speed at which the processes involving the decomposition of this organic matter can occur 3 to 4 times faster (Torres et al., 2021; Silveira et al., 2022), as the temperature and soil moisture are adequate during throughout the year, due to successive cultivation of crops, when compared to areas cultivated under natural conditions, as a consequence, it will also cause changes in soil attributes more quickly.
The traffic of agricultural machines in soils with humidity above the field capacity causes more problems with respect to compaction, while in drier soils this problem is less.
- 2.4.1. Mechanical resistance of soil to root penetration. All soil sampling and evaluations were carried out at soil depths of 0-0.1, 0.1-0.2, 0.2-0.3, and 0.3-0.4 m before planting the vegetables. Why did you choose such depths?
Several studies published in the international literature show that, in general, soil compaction always occurs in the layer around 30 cm in depth, that below 40 cm there is no change in physical attributes due to anthropic actions.
- 2.1.3. Soil type. Line 103: …and 19 g dm−3 of soil organic matter. The physical properties of soils are also influenced by the organic matter content of the soil. You have a wide variety of crops in the experiment, different sampling depths, two types of soils. There is no information on organic matter.
There are not two types of soil in the study, what exists is a soil with two classifications, the one described by Santos et al. (2018) (Oxisol) and the international (Hapludox Tipico) from the Soil Survey Staff (2014), however only one name was left in the text
The organic matter production via cover crops and relationships with soil attributes are highlighted in this study in various parts of the text. The last paragraph of the literature review deals specifically with organic matter, since the continuous contribution that occurs in the system increases the biodiversity of soil organisms, which in turn accelerates all processes involving organic matter in the soil, changing the physical attributes, which is confirmed later in this study and also discussed in the text, with citation of the studies compared to the confirmation of results by Cunha et al. (2011), Loss et al. (2012; 2019), Santos et al. (2014), Mottin et al. (2018).
- Perhaps the conclusions need to be described in more detail.
The conclusions have been reformulated
Yours sincerely
José Luiz Rodrigues Torres
corresponding author

Round 2
Reviewer 1 Report
I think that authors properly answered all questions and paper can be published as it is now.